# A Comparison of the Composition and Contamination of Soybean Cultivated in Europe and Limitation of Raw Soy Seed Content in Weaned Pigs’ Diets

**DOI:** 10.3390/ani10111972

**Published:** 2020-10-27

**Authors:** Anita Zaworska-Zakrzewska, Małgorzata Kasprowicz-Potocka, Magdalena Twarużek, Robert Kosicki, Jan Grajewski, Zuzanna Wiśniewska, Andrzej Rutkowski

**Affiliations:** 1Department of Animal Nutrition, Faculty of Veterinary Medicine and Animal Science, Poznan University of Life Sciences, 60-637 Poznan, Poland; anita.zaworska-zakrzewska@up.poznan.pl (A.Z.-Z.); zuzanna.wisniewska@up.poznan.pl (Z.W.); andrzej.rutkowski@up.poznan.pl (A.R.); 2Department of Physiology and Toxicology, Faculty of Biological Sciences, Kazimierz Wielki University, 85-064 Bydgoszcz, Poland; twarmag@ukw.edu.pl (M.T.); robkos@ukw.edu.pl (R.K.); jangra@ukw.edu.pl (J.G.)

**Keywords:** soybean seeds, varieties, mycotoxins, antinutrients, piglets, performance

## Abstract

**Simple Summary:**

Soy is the major source of protein in animal feeds worldwide. In Europe, only GMO-free varieties may be cultivated. Their chemical composition, contamination by fungi and yeast and the acceptable level in the diet of pigs have not been fully determined yet. This work comprised extensive analyses, both chemical (the composition of amino acids and anti-nutritional factors) and microbiological, also including mycotoxins. Moreover, digestibility and performance parameters were studied in 48 male post-weaning piglets for 28 days using diets, in which soybean meal was replaced by 0%, 5%, 10%, 15%, 20% and 25% soybean seed addition. The chemical composition of soybean seeds differed in terms of crude protein, ether extract, neutral detergent fibre and antinutrient contents. Seeds were also contaminated (but to varying intensity, which may have been influenced by the weather conditions during the seed harvest period) with fungi, yeast and mycotoxins, mainly zearalenone and deoxynivalenol. The digestibility coefficients of crude protein and dry matter in the diet were similar. Pigs‘ performance parameters were reduced strongly with increasing amounts of raw seeds in their diets, so a 5% of raw soy seed supplementation in pigs’ diet is recommended.

**Abstract:**

The aim of this study was to compare the chemical composition of European soy seeds. A mycological and toxigenic screening was carried out on 18 varieties of soy seeds harvested in Poland. Moreover, the level of soybean meal (SBM) substitution by raw soybean seeds was analysed in terms of its effect on young pigs’ performance (body weight gain, feed intake, feed utilisation) along with apparent total tract digestibility (ATTD) of dry matter and crude protein in the diets. In a 28-day trial, 48 male pigs were tested using a marker method with TiO_2_. In their diets, SBM was replaced by soy seeds in the amounts of 0%, 5%, 10%, 15%, 20% and 25%. In the last 3 days of the experiment, samples of excreta from each animal separately were collected three times per day. The chemical composition of soybean seeds differed in terms of their contents of crude protein, ether extract, neutral detergent fibre and raffinose family oligosaccharides, as well as the trypsin inhibitor activity. Seeds were also contaminated with fungi, yeast and mycotoxins, mainly zearalenone and deoxynivalenol. The ATTD of crude protein ranged from 70.6% to 77.6% and that of dry matter from 93.5% to 94.6%, with no differences between the groups being found (*p* > 0.05). Pigs’ performance parameters were reduced strongly with increasing amounts of raw seeds in the diets (*p* < 0.05). The results indicate that only a 5% addition of raw soy seeds in pigs‘ diet is recommended.

## 1. Introduction

There is increasing concern related to protein imports to Europe. Increasing the EU protein crop production enhances the possibilities for crop rotation, thus reducing the risk of crop diseases and stabilising EU farmers’ income. Moreover, socially desirable cultivation (e.g., non-GMO soybean production) can be successfully promoted [1]. Interest in soy cultivation has increased in recent years, and in 2017, European production reached over 2 thousand tonnes [2]. Every year, the number of soybean varieties is growing. Currently, there are at least 502 registered soybean varieties in the EU Plant variety database [3]. Such a number of varieties may indicate their variability, in particular connected with the yielding potential, but also the length of the growing season and their chemical composition. It needs to be remembered that soybean seeds are characterised by high contents of protein, fat, macro- and microelements [4]. However, the use of raw soybean seeds in pig nutrition is limited due to the high contents of anti-nutritive factors (ANFs) such as protease inhibitors, tannins, phytate, mycotoxins or allergens [5,6]. Raw soybean seed contents in diets differ within technological groups (piglets, fatteners or broiler chickens), typically not exceeding 5 to 10% [7,8,9,10]. To date, European soy varieties have not been considered in this context. It is hypothesised that due to the different background of individual varieties, seeds of cultivated soy will have a variable composition. The higher levels of raw seeds in pig diets may decrease nutrient digestibility and animal growth [11,12]. In view of the above, the aims of the present investigation were the following: (1) to analyse the chemical composition of seeds of 18 soybean varieties available in the EU (Poland); (2) to evaluate the impact of increasing levels of raw soybean seeds on rearing results of piglets; and (3) to determine the effect of increasing levels of raw seeds of soy cultivar (cv.) Augusta on the apparent total tract digestibility coefficients of dry matter and crude protein.

## 2. Materials and Methods

### 2.1. Soybean Seeds

Seeds of 18 soybean (*Glycine max* L. Merill) cvs.: Abelina, Aldana, Aligator, Annushka, Augusta, Brunensis, Erica, Lissabon, Madlen, Mavka, Merlin, Paradis, Petrina, Protina, Silesia, Sirelia, Solena and Naya were subjected to chemical and nutritional analyses. The data on soybean seeds were obtained from crops harvested in different years: Abelina, Aldana, Augusta, Mavka (2015–2017); Aligator, Annushka, Madlen (2015–2016); Paradis (2016); Erica, Merlin, Petrina (2016–2017) and Brunensis, Lissabon, Protina, Silesia, Sirelia, Solena, Naya (2017). Soy seeds came from different plant stations, i.e., Agro Youmis, Saaten Union, Saatbau, Danko, Prograin ZIA, EURALIS, the Poznan University of Life Sciences and the HR Strzelce IHAR Group. Soybean seeds of cv. Augusta for the experiment on pigs were obtained from an Experimental Station of the Poznan University of Life Sciences (Poland), which was established in 2002. Soybean cv. Augusta is a model variety in field cultivation and is considered to be the most suitable for the climatic conditions of Poland, giving a stable yield in various weather conditions. Seeds of this cultivar are characterised by the chemical composition closest to the cultivar’s average with the lowest urease activity, which is beneficial for animals.

### 2.2. Ethic Statement

All the experimental procedures complied with the guidelines of the Local Ethical Committee for Experiments on Animals in Poznan regarding animal experimentation and animal care under study (EU Directive 2010/63/EU for animal experiments). The pigs received all the necessary veterinary vaccinations and had unlimited access to water and feed.

### 2.3. Animal, Diets and Experimental Design

For the feeding study in pigs using soybean seeds cv., Augusta was selected because its chemical composition, levels of essential amino acids, protein and anti-nutritional factors were closest to the average for this crop, while urease activity was markedly lower than in seeds of other soybean varieties, which is one of the most relevant quality aspects of soybean seeds for monogastric animals. The experiment was conducted on 48 crossbreed male piglets (Naïma × (Pietrain × Duroc)) with an initial body weight of about 9.0 ± 0.2 kg aged 42 days. Pigs were randomly allocated to six dietary treatments (of 8 animals per group) and were kept in individual cages. The cages met the welfare requirements for pigs in accordance with the Polish legal guidelines applicable during the time of the experiments [13]. The control group (S0) was fed a diet with soybean meal (SBM), whereas the experimental diets contained 5% (S5), 10% (S10), 15% (S15), 20% (S20) and 25% (S25) raw soybean seeds (cv. Augusta) as a partial substitution of SBM. Rapeseed oil was added to the experimental diets to make the diets isocaloric. All the diets were provided in the mash form and prepared according to the Recommended Allowances and Nutritive Value of Feedstuffs for Swine [14] at the Experimental Animal Nutrition Station in Gorzyń, belonging to the Poznan University of Life Sciences. The chemical composition of the diets is presented in Table 1. To determine digestibility coefficients, 0.3% titanium dioxide was included in the diets as a marker. The experiment lasted 28 days. The average daily feed intake (ADFI) and average daily gain (ADG) were recorded, and the feed conversion ratio (FCR) was calculated. Data on ADFI were collected on day 1, day 14 and day 28 of the experiment. Individual observations and measurements were recorded for each animal using a weighing specialist veterinary platform ADE EHR4 (Poland). Based on the recorded data, the feed conversion ratio (FCR) was calculated.

### 2.4. Apparent Total Tract Digestibility

In the last 3 days of the trial, excreta from each animal were collected to plastic bags three times a day and frozen (−20 °C). The fresh faeces were collected from the straw floor in pens, and the unwanted straw waste was removed. After defrosting, the samples were mixed separately, and fresh dry matter was analysed in duplicate. The rest of excreta was lyophilised, and dry matter, titanium oxide and protein contents were analysed. The coefficients of apparent total tract digestibility (ATTD) of the components in the experimental feeds were calculated using the following formula proposed by Adeola [15] (Eq. 40.2):

The ATTD of the tested feedstuffs (ATTDf) was calculated using the following equation:ATTDf [%] = 100 − [(ID/AF)/(AD/IF))/100](1)
where ID is the marker concentration in the diet [g/kg DM]; AF is the nutrient content in faeces [g/kg DM]; AD is the nutrient content in the diet [g/kg DM]; and IF is the marker concentration in faeces [g/kg DM].

### 2.5. Chemical Analysis

For chemical analyses, all the samples were ground to pass through a 1.0- or 0.5-mm mesh size sieve. The seeds were analysed in duplicate for dry matter (DM), crude protein (CP), ether extract (EE), crude fibre (CF), crude ash (CA), acid detergent fibre (ADF), neutral detergent fibre (NDF) and total phosphorus contents using methods 934.01, 976.05, 920.39, 978.10, 942.05, 973.18, 984.27, and 965.17, respectively, according to AOAC [16]. Metabolic energy was calculated according to GfE [17]. The true protein (TP) content was determined according to method 934.01 [16], followed by suspension of protein with 20% triacetic acid. The amino acid content was determined with an AAA-400 Automatic Amino Acid Analyzer (Prague, Czech Republic) using ninhydrin (Sigma-Aldrich, Munich, Germany) for post-column derivatisation. Before analysis, the samples were hydrolysed with 6 N HCl (POCH, Gliwice, Poland) for 24 h at 110 °C [16]. Raffinose family oligosaccharides (RFOs) were extracted and assayed in a high-resolution GC 2010 gas chromatograph (Shimadzu, Japan). [18]. At first, carbohydrates were extracted from the pulverised material (40–45 mg each, in 3 replications) with 50% aqueous ethanol solution containing xylitol as an internal standard. After heating at 90 °C for 30 min, samples were centrifuged, and homogenate was transferred to the centrifuge filter. A portion of the filtrate was concentrated in a rotary evaporator until dry. Dry residues were derivatised with a mixture of trimethylsilyl-imidazole and pyridine at 80 °C for 45 min. Oligosaccharides were quantified using original standards purchased from Sigma-Aldrich (USA). Chromatograms were analysed with an integrator in the CHROMA 3.2 application (Pol-Lab, Wilkowice, Poland). The content of the assayed carbohydrates was calculated applying the internal standard method. Phytate-P was determined using the method described in [19], where phytic acid is precipitated with an acidic iron-III-solution of known iron content. The decrease in iron contents in the supernatant is a measure of the phytic acid content. The solutions were analysed using a UV spectrophotometer and measuring absorbance at 519 nm (Spektrofotometr Marcel Media, Poland). Trypsin inhibitor activity (TIA) was analysed according to PN EN ISO 14902:2005 [20]. This standard specifies a method for the determination of the TIA in soya products. This trypsin inhibitor activity is indicative of the degree of toasting in these products. The detection limit of the method is 0.5 mg/g. Trypsin inhibitors are extracted from the sample at pH 9.5. The residual trypsin activity is measured by adding benzoyl-L-arginine-*p*-nitroanilide as a substrate. The amount of released p-nitroaniline is measured spectrophotometrically (Spektrofotometr Marcel Media, Zielonka, Poland). Urease activity in soy products was determined by potentiometric titration [21]. The urease activity is determined by assaying the amount of ammonium nitrogen released from the urea solution, expressed per 1 g of product per minute at 30 °C. Additionally, soybeans must be degreased prior to analysis. In feed and faecal samples, titanium dioxide was assayed according to [22], where freeze-dried digesta samples were ashed and dissolved in 7.4 M sulphuric acid. Hydrogen peroxide (30% vol.) was subsequently added, resulting in the typical orange colour, the intensity of which was dependent upon the titanium concentration. The solutions were analysed using a UV spectrophotometer and measuring the absorbance at 410 nm (Spektrofotometr Marcel Media, Zielonka, Poland).

### 2.6. Mycotoxin Analysis

#### 2.6.1. Aflatoxins

##### Sample Preparation

The amount of 25 g of the sample was mixed with 2.5 g of NaCl and homogenised with 50 mL of MeOH:H_2_O (80:20) for 1 min. Then, the extract was filtrated and 10 mL were added to 40 mL of H_2_O, filtrated again, and subsequently 10 mL of the diluted extract were passed through an AflaTest^®^ column (Vicam, Watertown, MA, USA). The column was washed with two batches of 10 mL H_2_O. Toxins were eluted using 1 mL of MeOH to a sample vial and diluted with 1 mL of H_2_O. The whole sample was vortexed.

#### 2.6.2. Chromatographic Analysis

Aflatoxins were determined using HPLC with post-column derivatisation and fluorescence detection. HPLC: pump L-7100; autosampler L-7250; oven L-7300; FLD L-7480 (Merck-Hitachi, Darmstadt, Germany); chromatographic column: LiChroCART 250–4, LiChrospher 100 RP-18 (250 × 4 mm, 5 µm); mobile phase: ACN:MeOH:H_2_O (20:20:60) + 119 mg KBr + 100 ul 65%HNO_3_; flow rate: 1 mL/min; injection volume: 50 µL. Detection and quantification limits: AFB1: 0.015 ng/g and 0.05 ng/g; AFB2: 0.02 ng/g and 0.08 ng/g; AFG1: 0.25 ng/g and 0.75 ng/; and AFG2: 0.08 ng/g and 0.24 ng/g.

#### 2.6.3. Ochratoxin A

##### Sample Preparation

The amount of 25 g of the sample was homogenised for 2 min with 100 mL of ACN:H_2_O (60:40). The extract was filtrated and 5 mL of the supernatant were mixed with 55 mL of PBS solution; next, the mixture was filtrated again, and 48 mL of the diluted extract were applied to the OCHRAPREP^®^ column (Rhone Diagnostic Technologies Ltd., Glasgow, UK). The column was washed with 20mL of H_2_O. Ochratoxin A was eluted using 1.5 mL of MeOH:CH_3_COOH (98:2) to a sample vial. Then 1.5 mL of H_2_O was passed through the column. The sample was vortexed.

#### 2.6.4. Chromatographic Analysis

Ochratoxin A content was determined using HPLC with fluorescence detection. HPLC: LaChrom ELITE (Merck-Hitachi, Darmstadt, Germany), chromatographic column: LiChrospher 100 RP-18 (250 × 4 mm, 5 µm), mobile phase: ACN:2% CH_3_COOH (70:30), flow rate: 1 mL/min, injection volume: 50 µL. Detection and quantification limits: 0.13 ng/g and 0.40 ng/g, respectively.

#### 2.6.5. Trichothecenes and Zearalenone

##### Sample Preparation

A 12.5 g sample was homogenised for 3 min with 50 mL of ACN:H_2_O (80:20). To 4 mL of the filtrated extract, 40 µL of 13C-zearalenone solution (c = 1000 ng/mL) was added, and the mixture was passed through a Bond Elut^®^ Mycotoxin column (Agilent, Santa Clara, CA, USA). The amount of 50 µL of the internal standard solution (13C-DON (c = 2500 ng/mL),13C-T2 (c = 250 ng/mL) and 13C-HT2 (c = 250 ng/mL)) was added to 2 mL of the cleaned extract, and the mixture was evaporated to dryness with nitrogen. The dry residue was reconstituted in 495 µL of MeOH:H_2_O 1:4. The whole sample was vortexed.

#### 2.6.6. Chromatographic Analysis

Contents of trichothecenes and zearalenone were determined using HPLC with MS/MS detection. HPLC: Shimadzu Nexerra, mass spectrometer: API4000 (AB Sciex, Foster City, CA, USA); chromatographic column: Gemini C18NX (150 × 4.6 mm, 3 µm) (Phenomenex Inc., Torrance, CA, USA); mobile phase: A: H_2_O + 5 mM CH_3_COONH_4_ + 1% CH_3_COOH, B: MeOH + 5 mM CH_3_COONH_4_ + 1% CH_3_COOH; flow rate: 0.5 mL/min; injection volume: 7 µL. Detection and quantification limits: deoxynivalenol (DON): 1.0 ng/g and 3.0 ng/g; nivalenol (NIV): 1.0 ng/g and 3.0 ng/g; diacetoxyscirpenol (DAS): 0.33 ng/g and 1.0 ng/g; T-2 toxin: 0.2 ng/g and 0.6 ng/g; HT-2 toxin: 0.7 ng/g and 2.0 ng/g; and ZEA: 0.07 ng/g and 0.2 ng/g.

#### 2.6.7. Mycological Analysis

Quantitative analysis consisted of assessing the total count of fungi, i.e., the number of colony forming units (cfu) in 1 g of a given material [cfu/g]. Soybean samples were ground, and an analytical sample of 20g 20 ± 0.2 g was prepared. The material was suspended in 180 ± 2% ml of sterile dilution liquid, prepared according to PN EN ISO 6887–1 (July 2000), pH 7.0 ± 0.2 and homogenised in a microbiological homogeniser for 90 s. The total count was determined according to PN ISO 7954 with the authors’ modification (surface culture of 1 mL and 0.1 mL in triplicate). A series of ten-fold dilutions in sterile dilution fluid was made from the homogenised stock suspension. Surface culture according to Koch was made on microbiological YGC (yeast extract, glucose, chloramphenicol) of the following composition: yeast extract—5 g, glucose—20 g, chloramphenicol—0.1 g, agar—15 g, distilled water—1000 mL, and pH 6.6. Incubation was carried out for 5–7 days at 25 °C ± 1 °C. After the designated incubation time, colonies from dishes were counted, with the number of colonies ranging from 10 to 100. Initial separation into moulds and yeasts was made on the basis of microscopic preparations. Lactophenol microscopic preparations according to Amann were made from colonies grown on the YGC medium. Based on colony morphology and the type of sporulation, the dominant types of mould fungi were identified. Calculations were made [cfu/g] for each of the mould types.

### 2.7. Statistical Analysis

The data provided by seed analyses were presented as mean values± standard deviation (SD) calculated for each nutrient. The significance of differences between the groups in the experiment on pigs was calculated using one-way ANOVA with Duncan’s post hoc test, while an alpha level of *p* < 0.05 was used to assess the significance of differences between means. The obtained results were analysed statistically by calculating the arithmetic mean and ± SD for each characteristic. The statistical analysis was performed using SAS ver. 5.0. software (IO, Cary, USA).

## 3. Results

### 3.1. The Chemical Composition of Soybean Seeds

The chemical analyses, the essential amino acid profile and ANF contents of analysed soy seeds are presented in Table 2, Table 3 and Table 4. For these parameters, soybean varieties were grouped depending on the CP content in the seeds (low up to 37%, medium 37–40, high over 40% in DM). The mean dry matter content was about 91%. To compare the chemical characteristics of seeds from different varieties, nutrients were expressed as % of DM. Crude protein content in DM varied considerably from 33.44% in cv. Erica to 48.16% in cv. Protina. The mean CP value was 38.8% in dry matter. True protein content reached about 28.8% of DM. Crude fibre content ranged from 4.81 to 8.15% DM in seeds of cv. Silesia and Annushka. The mean fat content was 20% DM, with the lowest level of 16.97% noted in cv. Annushka and the highest in cv. Petrina and Solena at 23.9 and 23.7% DM, respectively. Mean crude ash content did not exceed 6.4% DM. The mean contents of ADF and NDF amounted to 8.3 and 10.4% DM, respectively. Calculated metabolic energy content ranged from 15.5 MJ/kg DM in cv. Erica and Lissabon to more than 17 MJ/kg in seeds of Abelina, Aligator, Madlen, Sirelia and Solena, with a mean value of approx. 16.7 MJ/kg DM. The essential amino acid profile of protein was similar for all the varieties. The average lysine content was 7.15 g/100 g of protein, whereas that of methionine ranged from 0.87 g/100 g of protein (Paradis) to 1.3 g/100 g of protein (Brunensis, Naya, Silesia), with a mean value of approx. 1.12 g/100 g of protein. The content of RFOs ranged from about 48–49 g/kg (Brunensis, Naya and Silesia) to about 74 g/kg DM (Merlin). The dominant oligosaccharide was stachyose (about 80% of total RFOs), followed by raffinose (approx. 15.7% of total RFOs) and verbascose (about 3.9% of total RFOs). Mean trypsin inhibitor activity was 29.8 TIA mg/g DM and ranged from 15.70 mg/g (Sirelia) to 38.52 mg/g (Annushka). In turn, P-phytate content varied considerably, ranging from 0.3 to 0.62% DM, with a mean value of about 0.46% DM. Phytate phosphorus constituted approx. 60.3% of total phosphorus content. Urease activity was measured only in some samples. These values ranged from 2.27 (Augusta) to 4.94 IU/g DM (Sirelia) with a mean value of about 3.89 IU/g DM.

Among the assayed mycotoxins (Table 5), ZEN and DON were the most common, as ZEN was detected in 24 samples (75%) with a mean concentration of 42.1 μg/kg, while DON was detected in 22 samples (68.75%) at a mean concentration of 23.6 μg/kg. The maximum ZEN level was recorded in cv. Aligator in 2016 (529 μg/kg). The maximum DON content was also recorded in cv. Aligator in 2016 (244 μg/kg). A total of 31.25% of the analysed samples contained the T-2 toxin with a mean concentration of 19.6 μg/kg. The maximum concentration of this mycotoxin was recorded in cv. Brunensis in 2017 (181 μg/kg). In turn, 28.12% of the analysed samples contained the HT-2 toxin, with a mean concentration of 70.58 μg/kg. Similarly as in the case of T2, the maximum concentration of this mycotoxin was also assayed in cv. Brunensis in 2017 (288 μg/kg). Ochratoxin A was detected only in one sample in the crop of 2016 (maximum concentration of 1.32 µg/kg—cv. Abelina). Similarly, NIV and DAS were also detected only in one sample in the same year of harvest in cv. Mavka (maximum concentration of 4.29 µg/kg and below the limit of quantification (1.0 μg/kg, respectively)). Aflatoxins were absent in all the samples. It was concluded that the analysed samples contained a majority of Fusarium metabolites, but their concentrations were low when compared to the maximum permissible level. The quantitative and qualitative composition of the total number of fungi and moulds are shown in Table 6. Samples of soybean seeds differed in the counts of fungi inhabiting them. In most soybean samples analysed, the total count of moulds was higher than that of yeast species. The number of mould species isolated from the samples ranged from 1 to 7. The qualitative composition of mould fungi varied significantly for each variety of the tested material. Most of the tested samples were infected simultaneously by yeasts, environmental fungi—both allergenic (*Alternaria*, *Cladosporium*, and *Mucor*) and pathogenic strains producing secondary metabolites, i.e., mycotoxins (*Aspergillus*, *Penicillium*, *Fusarium*, and *Eurotium*). Fungi with an isolation frequency above 20% represented the genera *Endomyces*, *Alternaria*, *Acremonium*, *Penicillium*, *Cladosporium*, *Fusarium* and *Phoma*. The analysed soybean seeds showed no visible signs of moulding; however, specific myco-biocenosis confirmed the colonisation of samples by many types of filamentous fungi.

### 3.2. Animal Experiment

No significant effect (*p* > 0.05) was observed for SBM substitution with crude soybean seeds in terms of ATTD coefficients of dry matter and crude protein (Table 7). In the diets with 20% and 25% contents of raw soybean seeds, higher ATTD of DM and CP were recorded. In the experimental groups, in which soybean seeds were supplemented at 5% and 10%, the one with the lowest ATTD levels was observed for DM and CP. In the whole experimental period, significant differences in animal growth were noticed between the dietary treatments (Table 8). The ADG with ADFI gradually decreased with an increase in raw soybean seed inclusion in the diet. Consequently, FCR increased proportionally with the substitution of SBM with raw soybean seeds. Replacement of SBM without reducing weaners’ performance was acceptable only at a 5% inclusion of raw soybean seeds. Substitution of 10%, 15%, 20%, and 25% SBM with raw seeds resulted in a deterioration of pigs’ performance.

## 4. Discussion

### 4.1. The Chemical Composition of Soybean Seeds

The chemical composition of soybean seeds may vary significantly depending on the variety and environmental conditions [23]. Additionally, seeds analysed in this study differed in terms of their chemical composition. The seeds contained on average 91.18% of dry matter, which is lower when compared to the value of 93% reported by Grela and Skomiał [14]. However, these results indicate that either the seeds had been dried or weather conditions during harvest were very good to provide high-quality material. The level of crude protein (38.88%) was definitely higher than 34.51% as reported by Grela and Skomiał [14] and 35.3% given by Jarecki and Bobrecka-Jamro [24]. In turn, Redondo-Cuenca et al. [25] found protein contents from 40.4% in DM for seeds from conventional cultivation. Pande et al. [26] and Jarecki and Bobrecka-Jamro [24] were of the opinion that protein contents of soybean seeds and other legume seeds depend not only on the genetic properties of the variety, but also climatic conditions prevalent during the growing season as well as agrotechnical factors, especially nitrogen fertilisation. According to Batista et al. [27], in addition to protein content, an important characteristic in terms of the quantitative profile is also connected with fat content in soybean seeds. Soybean oil is one of the most preferred vegetable oils used for food and feed components. Oil content ranges from 16.97% to 23.91%, with an average of 20.00%, which was similar to 20.5% as reported by Jarecki and Bobrecka-Jamro [24], and 21% given by Grela and Skomiał [14]. The above studies also showed similar levels of crude fibre, ash and metabolic energy for pigs. It has been widely documented that soybean seed amino acid composition varies depending on environmental factors, especially the location and temperature [28]. In the case of amino acid analysis, lysine, arginine, and threonine contents were observed to be higher than those reported by Grela and Skomiał [14], but similar to the data presented by Brzóska and Śliwa [29]. Soybean seeds contain many ANFs. The analysed seeds contained similar mean levels of RFOs, TIA and phytate-P to those reported by other authors [26], but lower than those given by Kołata [30,31] and Zaworska-Zakrzewska et al. [6]. It should be emphasised here that the analysed soybean varieties were not obtained from the same soil-climatic, agrotechnical and other comparable conditions (years); therefore, it is difficult to definitely state that the assayed chemical composition and nutritional value are strictly cultivar dependent.

Mycotoxin contamination of soybean is not considered a significant problem when compared to such commodities as maize, cottonseed, peanuts, barley and other grains. In the early surveys conducted by the U.S. Department of Agriculture in 1995, 1046 soybean samples collected from different regions of the United States were examined for aflatoxin contamination. Aflatoxin presence was confirmed at low levels (7–14 ppb) in only two of the tested samples [32]. In their study, Binder et al. [33] tested 122 soybean samples that came from Asia and the Pacific region. Aflatoxin was found in only 2% (maximum 13 ppb, median 9 ppb), zearalenone in 17% (maximum 1078 ppb, median 57 ppb), ochratoxin in 13% (maximum 11 ppb, median 7 ppb), and DON and fumonisins each in 7% of the analysed samples (DON: maximum 1347 ppb, median 264 ppb; fumonisins: maximum 331 ppb, median 154 ppb). Additionally, in our study, DON, T2, HT2 and ZEN were detected above the threshold in 28.13%, 9.38% 25% and 50% of samples, respectively. The results obtained in our research correlated with the studies of Rodrigues and Naehrer [34] and Zaworska-Zakrzewska et al. [6] showing that DON and ZEN are the mycotoxins most frequently found in soybean. Co-occurrence of at least two mycotoxins was observed in 62.5% of cases. Gutleb et al. [35] also reported in soybean seeds the occurrence of at least two toxins in 72% cases. The investigated microflora of a subset of soy seeds (*n* = 32) suggested that *Fusarium* spp., *Alternaria* spp., *Aspergillus* spp., *Cladosporium* spp. and *Acremonium* spp. frequently colonise soy seeds; however, in soybeans, *Alternaria* was the dominant genus, which can produce mycotoxins. They may be found on various materials, in house and warehouse dust, as well as wallpaper or wood. They can cause numerous allergies. They can be found in plant products, in which toxins such as alternariol and tenuazonic acid can be detected [36]. Similarly, Janda and Wolska [37], when studying the qualitative and quantitative composition of fungi present in soybeans in 15 samples taken from the retail chains, isolated 95 strains belonging to 40 species. The dominant species were *Penicillium chrysogenum* and *Eurotium herbariorum*, which belong to storage moulds and produce toxic secondary metabolites. The obtained results indicate that mould fungi and mycotoxins were found with varying intensity depending on the seed variety and year of cultivation, which may be due to non-optimal storing conditions of the feed. They can therefore be a potential source of microbial contamination in food and feed.

### 4.2. Animal Experiment

The increasing raw soy seed contents in pig diets did not affect apparent digestibility of dry matter and crude protein. Grela and Skomiał [14] presented apparent digestibility coefficients for DM and CP of raw soy seeds at 85 and 89%, respectively, but the actual effect of seed levels in the diet was not recognised. On the other hand, digestibility of nutrients in raw soy seeds may be the result of inhibitors and lectin contents. Tagliapietra et al. [38] claimed that soybean seeds with a relatively high Kunitz inhibitor activity may be characterised by lower digestibility; however, digestibility depends mainly on the activity of inhibitors, rather than their amount. Additionally, Herkelman et al. [39] pointed out that apparent digestibility increased when animals were fed improved soybean seeds, which contain lower amounts of inhibitors than conventional varieties. Moreover, crop varieties can have a significant influence on feed use [40]. In the current study, the level of 5% raw seeds did not disturb performance results, whereas a greater addition of raw seeds in pig diets negatively affected the pigs’ performance. It is clear that this is not directly the result of diet digestibility, but it is rather connected with higher contents of antinutrients such as lectins, phytate, inhibitors, urease as well as mycotoxins, which may have a negative impact on diet palatability, thus reducing the utilisation of some essential nutrients. This is proved by a lower feed intake and lowed FCR of animals offered more than 5% raw soy seeds in their diet. Thus, the alterations in the reduced pigs’ performance must be attributed to a lower feed intake due to reduced palatability. Zaworska-Zakrzewska et al. [6] replaced 5% of the SBM protein with raw soybean seed protein in the diet of fatteners. This level showed no negative response in terms of growing animals’ performance. The ADG, AFI and FCR were comparable, regardless of the diet composition. However, results of Young et al. [12] showed that low levels of raw soybean seeds in the diets of growing pigs (5–10%) and the resulting low anti-nutrient contents did deteriorate feed quality, palatability and probably also its digestibility. Thus, this could be the result of the specific characteristics of individual soy cultivars and especially the content of mycotoxins. The seeds of cv. Augusta used in the experiment contained 19.8 ppb DON and 2.57 ppb ZEN, which could negatively impact AFI and FCR. Pierron et al. [41] and Reddy et al. [42] found that contamination of feed by DON and ZEN reduces FI, ADWG and FCR. It is consistent with the results of this study, where the diet with increasing soy levels, as a consequence of higher mycotoxin contents, significantly reduced piglet performance parameters. This can be important information for farmers who should analyse feed mixtures containing soy for the presence of fungal contamination and mycotoxins.

## 5. Conclusions

The chemical composition of soy seeds of different varieties collected in 2015, 2016 and 2017 in Poland differed especially in terms of crude protein, ether extract, neutral detergent fibre, the raffinose family oligosaccharides and trypsin inhibitor activity. All the seeds contained from 1 to 7 different mould and yeast species. Seeds were also contaminated by mycotoxins, mainly zearalenone and deoxynivalenol. Total tract digestibility of dry matter and protein was not affected regardless of the proportion of raw seeds in the feed. Replacement of soybean meal with raw soybean seeds in the diet for young pigs in the amount of 5% had generally no negative effect on the animals’ performance. However, higher raw seed levels (10–25% in diet) resulted in a significantly lower feed intake and utilisation as well as reduced growth of pigs.

## Figures and Tables

**Table 1 animals-10-01972-t001:** Composition and nutritional value of feed mixtures.

Components (%)	S0	S5	S10	S15	S20	S25
Raw soybean seeds (35.2% CP)	0.00	5.00	10.00	15.00	20.00	25.00
Soybean meal (46% CP)	25.00	21.50	17.50	13.50	10.00	6.00
Wheat	47.70	46.79	46.78	46.87	45.86	45.34
Corn	20.00	20.00	20.00	20.00	20.00	20.00
Monocalcium phosphate	1.20	1.10	1.10	1.00	1.00	1.00
Limestone	1.30	1.30	1.30	1.30	1.30	1.30
Salt	0.34	0.34	0.340	0.340	0.340	0.340
Rapeseed oil	3.50	3.00	2.00	1.00	0.50	0.00
Premix *	0.50	0.50	0.50	0.50	0.50	0.50
L-lysine	0.14	0.15	0.15	0.16	0.17	0.18
DL-methionine	0.02	0.02	0.03	0.03	0.03	0.03
Titanium oxide	0.30	0.30	0.30	0.30	0.30	0.30
Nutritional value						
EM (calculated) MJ/kg	14.10	14.20	14.10	14.10	14.10	14.20
Crude protein (g/kg)	185.00	186.00	186.00	185.00	186.00	185.00
Lysine (g/kg)	10.60	10.70	10.60	10.70	10.60	10.70
Methionine + cystine (g/kg)	6.30	6.30	6.30	6.30	6.30	6.30
Tryptophane (g/kg)	2.30	2.30	2.30	2.30	2.20	2.20
Threonine (g/kg)	6.90	6.90	6.80	6.70	6.50	6.60
Ca (g/kg)	9.60	9.40	9.40	9.20	9.20	9.20
P (g/kg)	6.60	6.40	6.50	6.40	6.50	6.50
Na (g/kg)	1.50	1.50	1.50	1.50	1.50	1.5
Crude fibre (g/kg)	35.10	35.60	36.10	36.60	37.10	37.40

* Mineral and vitamin premix content per 1 kg feed. Fe (75 mg); Cu (20 mg); Co (0.3 mg); Mn (30 mg); Zn (75 mg); I (0.6 mg); Se (0.15 mg); vitamin A 7500 IU; vitamin D3, 1500 IU; vitamin E, 52.5 mg; vitamin K3, 1.1 mg; vitamin B1, 1.1 mg; vitamin B2, 3.0 mg; vitamin B6, 2.25 mg; choline chloride, 200 mg; pantothenic acid, 7.5 mg; nicotinic acid, 15 mg; folic acid; 1.5 mg; vitamin B12, 18.5 µg, biotin; 75 µg; Ca, 1.3 g; antioxidants (butylated hydroxyanisole, butylated hydroxytoluene), CP—crude protein.

**Table 2 animals-10-01972-t002:** The average chemical composition of different soybean seeds varieties (% in dry matter (DM)) with a low level (about 37%) of protein in DM.

Soybean Variety		Erica	Petrina	Annushka	Brunensis
Dry matter (%)		92.46 ± 0.43	91.13 ± 1.66	91.48 ± 1.79	93.54 ± 1.12
True protein (%)		22.98 ± 2.17	25.63 ± 2.01	25.49 ± 0.80	27.18 ± 0.88
Crude protein (%)		33.44 ± 3.56	35.68 ± 1.65	36.30 ± 0.67	36.59 ± 0.58
Crude far (%)		7.21 ± 0.20	6.53 ± 0.58	8.14 ± 0.15	6.02 ± 0.18
Ether extract (%)		19.99 ± 1.05	23.91 ± 1.22	16.97 ± 0.10	19.66 ± 1.05
Crude ash (%)		6.34 ± 0.14	5.68 ± 0.35	5.89 ± 0.11	5.61 ± 1.11
Metabolic energy (MJ/kg)		15.59 ± 0.71	16.48 ± 0.44	16.67 ± 0.31	16.98 ± 0.58
Acid detergent fibre, (%)		9.25 ± 0.87	8.66 ± 0.42	9.36 ± 0.02	6.84 ± 0.47
Neutral detergent fibre, (%)		10.98 ± 1.19	10.00 ± 0.96	12.04 ± 0.56	9.48 ± 0.64
g/100 g Crude protein	Threonine	4.18 ± 0.06	4.15 ± 0.11	4.03 ± 0.01	3.87 ± 0.06
	Methionine	1.13 ± 0.23	1.15 ± 0.30	0.94 ± 0.03	1.34 ± 0.08
	Cysteine	1.49 ± 0.05	1.42 ± 0.26	1.47 ± 0.01	1.40 ± 0.11
	Valine	4.74 ± 0.05	4.77 ± 0.04	4.81 ± 0.11	4.51 ± 0.05
	Isoleucine	4.50 ± 0.09	4.47 ± 0.01	4.51 ± 0.12	4.23 ± 0.05
	Leucine	7.84 ± 0.04	7.79 ± 0.02	7.63 ± 0.20	7.30 ± 0.07
	Phenyl-alanine	4.85 ± 0.03	4.89 ± 0.18	5.01 ± 0.07	4.51 ± 0.05
	Histidine	3.20 ± 0.19	3.11 ± 0.03	3.18 ± 0.02	2.81 ± 0.17
	Lysine	7.39 ± 0.00	7.22 ± 0.34	7.12 ± 0.10	7.04 ± 0.08
	Arginine	8.22 ± 0.40	8.26 ± 0.09	8.75 ± 0.14	8.29 ± 0.11
Total content RFOs (g/kg)		61.88 ± 12.03	59.02 ± 12.84	66.43 ± 0.71	49.43 ± 1.25
% of sugars in RFOs	Raffinose	14.30 ± 0.13	15.77 ± 0.20	16.69 ± 2.78	20.14 ± 0.56
	Stachyose	81.43 ± 1.99	79.32 ± 1.23	80.94 ± 2.91	73.43 ± 2.11
	Verbascose	4.26 ± 1.85	4.89 ± 2.44	2.36 ± 0.12	6.41 ± 0.12
TIA (mg/g)		26.66 ± 1.86	25.66 ± 3.36	38.52 ± 5.07	18.90 ± 0.97
Urease activity (IU/g)		5.27 ± 0.06	4.75 ± 0.07	6.37 ± 0.05	4.10 ± 0.00
P-phyt (%)		0.51 ± 0.10	0.39 ± 0.01	0.47 ± 0.06	0.30 ± 0.12
P-phyt/P total (%)		63.00 ± 13.00	58.56 ± 3.72	59.56 ± 2.09	49.43 ± 0.13

RFOs—oligosaccharides from raffinose family, TIA—trypsin inhibitor activity, P-phyt—Phytate-P; results are expressed as means ± standard deviation.

**Table 3 animals-10-01972-t003:** The average chemical composition of different soybean seeds varieties (% in DM), with a medium level of protein (37 to 40% in DM).

Soybean Variety		Aldana	Solena	Mavka	Madlen	Abelina	Aligator	Merlin	Augusta	Naya
Dry matter (%)		94.53 ± 2.27	86.98 ± 0.96	90.17 ± 2.14	88.50 ± 0.73	90.12 ± 1.11	92.37 ± 1.33	90.71 ± 1.49	91.61 ± 1.76	93.64 ± 0.96
True protein (%)		26.85 ± 1.02	29.90 ± 1.17	28.24 ± 1.60	28.75 ± 1.38	27.94 ± 2.69	24.52 ± 2.94	30.05 ± 3.62	27.59 ± 3.91	28.64 ± 1.11
Crude protein (%)		37.31 ± 0.76	37.46 ± 1.14	37.52 ± 2.75	37.71 ± 1.86	37.83 ± 1.75	38.02 ± 0.16	38.05 ± 1.30	39.29 ± 2.78	39.52 ± 0.98
Crude far (%)		6.09 ± 0.56	6.00 ± 0.31	5.97 ± 0.43	6.46 ± 0.05	6.48 ± 0.22	7.10 ± 0.16	6.45 ± 0.51	7.30 ± 0.52	5.55 ± 0.36
Ether extract (%)		20.13 ± 2.91	23.72 ± 1.08	20.80 ± 2.73	17.77 ± 0.08	21.78 ± 1.12	19.85 ± 0.05	22.54 ± 4.00	19.35 ± 1.60	17.41 ± 1.09
Crude ash (%)		5.88 ± 0.24	5.75 ± 0.35	5.72 ± 0.44	5.95 ± 0.11	5.52 ± 0.26	6.11 ± 0.01	5.53 ± 0.03	6.14 ± 1.77	5.43 ± 0.55
Metabolic energy (MJ/kg)		16.69 ± 0.21	17.03 ± 0.47	16.86 ± 0.71	17.08 ± 0.37	17.06 ± 0.49	17.06 ± 0.71	16.33 ± 0.25	16.63 ± 0.62	16.82 ± 0.74
Acid detergent fibre (%)		8.56 ± 0.97	8.20 ± 0.74	8.62 ± 0.37	7.84 ± 0.63	8.38 ± 0.28	7.77 ± 1.00	8.08 ± 0.29	8.97 ± 0.81	6.80 ± 0.65
Neutral detergent fibre (%)		10.20 ± 0.94	10.05 ± 0.71	10.54 ± 0.26	11.68 ± 0.06	10.11 ± 0.27	10.28 ± 0.98	9.79 ± 0.45	11.24 ± 1.04	9.88 ± 0.88
g/100 g Crude protein	Threonine	4.07 ± 0.06	3.85 ± 0.10	3.93 ± 0.07	4.10 ± 0.10	4.04 ± 0.12	4.08 ± 0.18	3.90 ± 0.14	4.02 ± 0.18	3.94 ± 0.11
	Methionine	1.09 ± 0.17	1.25 ± 0.13	1.03 ± 0.12	0.94 ± 0.04	1.07 ± 0.15	0.95 ± 0.01	1.02 ± 0.16	1.09 ± 0.13	1.30 ± 0.13
	Cysteine	1.40 ± 0.04	1.27 ± 0.19	1.46 ± 0.10	1.44 ± 0.11	1.54 ± 0.09	1.44 ± 0.17	1.35 ± 0.05	1.47 ± 0.09	1.40 ± 0.06
	Valine	4.82 ± 0.03	4.89 ± 0.02	4.69 ± 0.08	5.04 ± 0.09	4.85 ± 0.04	4.89 ± 0.15	4.63 ± 0.07	4.82 ± 0.10	4.65 ± 0.08
	Isoleucine	4.55 ± 0.07	4.53 ± 0.18	4.41 ± 0.08	4.62 ± 0.15	4.54 ± 0.06	4.63 ± 0.07	4.39 ± 0.10	4.53 ± 0.09	4.42 ± 0.03
	Leucine	7.78 ± 0.18	7.84 ± 0.01	7.58 ± 0.17	7.78 ± 0.28	7.72 ± 0.08	7.88 ± 0.00	7.51 ± 0.20	7.57 ± 0.09	7.64 ± 0.16
	Phenylalanine	4.97 ± 0.12	4.94 ± 0.04	4.88 ± 0.24	5.18 ± 0.14	4.98 ± 0.13	5.14 ± 0.13	4.82 ± 0.20	4.90 ± 0.16	4.75 ± 0.06
	Histidine	3.10 ± 0.10	2.96 ± 0.16	3.06 ± 0.06	3.19 ± 0.07	3.11 ± 0.11	3.22 ± 0.12	2.94 ± 0.13	3.08 ± 0.21	2.91 ± 0.09
	Lysine	7.20 ± 0.10	7.43 ± 0.18	7.01 ± 0.04	7.28 ± 0.23	7.22 ± 0.12	7.19 ± 0.18	7.04 ± 0.14	7.32 ± 0.09	7.21 ± 0.05
	Arginine	8.65 ± 0.03	8.92 ± 0.12	8.54 ± 0.62	9.09 ± 0.21	8.76 ± 0.21	8.89 ± 0.13	8.31 ± 0.18	8.56 ± 0.16	8.80 ± 0.11
Total RFO content (g/kg)		56.03 ± 5.71	52.44 ± 2.17	60.56 ± 0.42	67.89 ± 5.98	58.92 ± 12.86	56.96 ± 9.09	74.15 ± 0.05	68.94 ± 3.59	49.55 ± 1.88
% of sugars in RFOs	Raffinose	22.03 ± 6.72	15.52 ± 0.50	14.04 ± 1.65	16.86 ± 1.86	16.70 ± 1.46	16.78 ± 0.14	12.66 ± 1.49	15.77 ± 1.67	15.15 ± 1.15
	Stachyose	73.17 ± 5.30	79.71 ± 1.44	82.40 ± 1.97	80.90 ± 1.80	78.72 ± 3.35	79.97 ± 0.13	84.05 ± 1.24	80.90 ± 1.73	78.30 ± 0.47
	Verbascose	4.79 ± 2.71	4.76 ± 0.73	3.55 ± 0.62	2.23 ± 0.06	4.57 ± 2.72	3.23 ± 0.27	3.27 ± 0.25	3.31 ± 0.18	6.53 ± 1.88
TIA (mg/g)		28.82 ± 9.54	18.5 ± 1.19	31.67 ± 9.86	28.48 ± 8.38	35.73 ± 16.52	37.06 ± 13.23	36.75 ± 20.39	35.09 ± 13.30	19.00 ± 2.12
Urease activity (IU/g)		5.56 ± 0.30	3.54 ± 0.87	5.08 ± 0.00	5.08 ± 0.04	4.64 ± 0.00	3.80 ± 0.10	6.00 ± 0.08	2.27 ± 0.12	3.87 ± 0.00
P-phyt(%)		0.46 ± 0.08	0.44 ± 0.16	0.47 ± 0.01	0.47 ± 0.03	0.42 ± 0.04	0.47 ± 0.02	0.53 ± 0.06	0.47 ± 0.07	0.38 ± 0.04
P-phyt/P total (%)		57.55 ± 5.03	54.29 ± 0.23	61.89 ± 8.03	59.22 ± 4.15	55.70 ± 3.29	61.89 ± 7.22	69.85 ± 2.10	61.37 ± 5.83	56.25 ± 0.04

RFOs—oligosaccharides from the raffinose family, TIA—trypsin inhibitor activity, P-phyt—Phytate-P; results are expressed as means ± standard deviation.

**Table 4 animals-10-01972-t004:** The average chemical composition of different soybean seeds varieties (% in DM) with a high level of protein (above 40% in DM).

Soybean Variety		Lissabon	Sirelia	Paradis	Silesia	Protina
Dry matter (%)		88.74 ± 1.24	89.48 ± 1.45	92.59 ± 1.44	93.51 ± 1.69	89.72 ± 0.87
True protein (%)		33.18 ± 1.78	33.18 ± 0.98	27.89 ± 1.54	32.30 ± 1.47	38.49 ± 1.22
Crude protein (%)		40.43 ± 1.58	40.87 ± 0.96	42.03 ± 1.98	43.68 ± 1.25	48.16 ± 1.35
Crude far (%)		6.12 ± 0.47	6.17 ± 0.29	6.20 ± 0.62	4.81 ± 0.18	6.35 ± 0.64
Ether extract (%)		22.99 ± 1.16	21.07 ± 1.41	17.58 ± 1.66	17.46 ± 1.25	17.04 ± 1.18
Crude ash (%)		6.06 ± 0.55	5.74 ± 0.28	5.26 ± 0.58	5.60 ± 0.41	6.01 ± 0.39
Metabolic energy (MJ/kg)		15.77 ± 0.41	17.06 ± 0.53	16.49 ± 0.38	16.71 ± 0.48	16.67 ± 0.54
Acid detergent fibre (%)		8.34 ± 0.59	6.96 ± 0.54	8.50 ± 0.91	9.47 ± 0.65	8.55 ± 0.81
Neutral detergent fibre (%)		10.16 ± 1.18	10.26 ± 0.86	11.19 ± 0.68	9.68 ± 0.71	9.80 ± 0.68
g/100 g Crude protein	Threonine	3.86 ± 0.06	3.53 ± 0.03	3.94 ± 0.06	3.80 ± 0.07	3.78 ± 0.12
	Methionine	1.24 ± 0.16	1.20 ± 0.11	0.87 ± 0.16	1.35 ± 0.10	1.23 ± 0.22
	Cysteine	1.37 ± 0.09	1.21 ± 0.14	1.24 ± 0.03	1.41 ± 0.18	1.34 ± 0.11
	Valine	4.80 ± 0.13	4.62 ± 0.03	4.82 ± 0.09	4.61 ± 0.09	4.64 ± 0.10
	Isoleucine	4.47 ± 0.05	4.36 ± 0.11	4.59 ± 0.04	4.38 ± 0.17	4.39 ± 0.07
	Leucine	7.82 ± 0.09	7.58 ± 0.12	7.92 ± 0.11	7.63 ± 0.16	7.67 ± 0.08
	Phenyl-alanine	4.88 ± 0.10	4.73 ± 0.03	5.17 ± 0.19	4.75 ± 0.10	4.89 ± 0.02
	Histidine	2.98 ± 0.11	2.86 ± 0.02	3.07 ± 0.03	2.88 ± 0.11	2.82 ± 0.13
	Lysine	7.26 ± 0.10	7.11 ± 0.07	7.05 ± 0.06	7.08 ± 0.05	7.08 ± 0.12
	Arginine	8.64 ± 0.17	8.54 ± 0.20	8.78 ± 0.13	8.92 ± 0.02	9.15 ± 0.04
Total RFO content (g/kg)		51.63 ± 2.23	55.02 ± 3.01	69.43 ± 0.98	48.91 ± 3.44	68.3 ± 1.99
% of sugars in RFOs	Raffinose	14.77 ± 0.82	13.35 ± 1.87	11.70 ± 1.11	17.54 ± 1.24	9.29 ± 0.72
	Stachyose	82.31 ± 1.64	83.42 ± 4.20	86.81 ± 1.60	75.85 ± 3.60	88.14 ± 2.05
	Verbascose	2.90 ± 1.52	3.21 ± 0.70	1.48 ± 0.24	6.60 ± 1.39	2.56 ± 1.22
TIA (mg/g)		18.4 ± 0.99	15.7 ± 0.99	25.29 ± 3.47	23.1 ± 2.00	20.80 ± 2.11
Urease activity (IU/g)		3.89 ± 0.00	4.94 ± 0.08	NA	4.24 ± 0.00	3.23 ± 0.01
P-phyt(%)		0.47 ± 0.00	0.45 ± 0.10	0.50 ± 0.01	0.40 ± 0.05	0.62 ± 0.07
P-phyt/P total (%)		59.15 ± 0.01	54.79 ± 0.15	68.66 ± 0.02	58.73 ± 0.10	68.29 ± 0.05

RFOs—oligosaccharides from the raffinose family, TIA—trypsin inhibitor activity, P-phyt—Phytate-P; results are expressed as means ± standard deviation.

**Table 5 animals-10-01972-t005:** Presence of mycotoxins in different varieties and harvest years of soybean seeds (ppb).

Soybean Variety	AF	OTA	DON	NIV	DAS	T2	HT2	ZEN	Year of Harvest
Abelina	0	0	0	0	0	0	0	<0.20	2015
	0	1.32	<3.00	0	0	0	0	0.94	2016
	0	0	<3.00	0	0	0	0	47.60	2017
Aldana	0	0	4.42	0	0	0	0	0.68	2015
	0	0	0	0	0	0	0	0	2016
	0	0	<3.00	0	0	0	0	<0.20	2017
Aligator	0	0	3.78	0	0	0	0	0.70	2015
	0	0	244	0	0	<0.60	3.91	529	2016
Annushka	0	0	0	0	0	0	0	0	2015
	0	0	<3.00	0	0	0	0	0	2016
Augusta	0	0	19.8	0	0	<0.60	0	2.57	2015
	0	0	0	0	0	0	0	0	2016
	0	0	4.24	0	0	0	0	3.04	2017
Brunensis	0	0	15.8	0	0	181	288	5.84	2017
Erica	0	0	0	0	0	0	0	0	2016
	0	0	<3.00	0	0	<0.60	3.63	<0.20	2017
Lissabon	0	0	3.09	0	0	0	0	<0.20	2017
Madlen	0	0	0	0	0	0	0	<0.20	2015
	0	0	<3.00	0	0	0	0	0	2016
Mavka	0	0	0	0	0	<0.60	274	<0.20	2015
	0	0	178	4.29	<1.00	8.15	52.4	396	2016
	0	0	0	0	0	0	0	0.46	2017
Merlin	0	0	0	0	0	0	0	0	2016
	0	0	<3.00	0	0	<0.60	3.63	<0.20	2017
Naya	0	0	7.33	0	0	3.07	4.74	19.3	2017
Paradis	0	0	<3.00	0	0	0	0	0	2016
Petrina	0	0	<3.00	0	0	0	0	0.22	2016
	0	0	<3.00	0	0	0	0	<0.20	2017
Protina	0	0	<3.00	0	0	<0.60	2.88	1.12	2017
Silesia	0	0	<3.00	0	0	<0.60	<2.00	0.68	2017
Sirelia	0	0	0	0	0	0	0	0.38	2017
Solena	0	0	<3.00	0	0	0	0	0.47	2017

Mycotoxins: AF—aflatoxin, OTA—ochratoxin, DON—deoxynivalenol, NIV—nivalenol, DAS—diacetoxyscirpenol, T2—toxin from the trichothecene group, HT2—toxin HT2, ZEN—zearalenone.

**Table 6 animals-10-01972-t006:** Contents of mould groups in selected analysed soy seeds.

Soybean Variety	Total (CFU/g)	Total Mould Count (CFU/g)	Total Yeast Count (CFU/g)	% Content of Mould
Abelina	<100	<50	<50	*Alternaria, Cladosporium, Eurotium, Acremoniu*m
Aldana	3.2 × 10^2^	1.3 × 10^2^	1.9 × 10^2^	68% *Endomyces*, 14% *Alternaria*, 7% *Aspergillus*, 7% *Cladosporium*, 4% *Curvularia*
Augusta	2.1 × 10^2^	2.0 × 10^2^	<20	45% *Alternaria*, 29% *Acremonium*, 11% *Penicillium*, 5% *Rhizopus*, 5% *Cladosporium*, 2.5% *Aspergillus,* 2.5% *Scopulariopsis*
Brunensis	2.1 × 10^3^	1.0 × 10^3^	1.1 × 10^3^	59% *Alternaria,* 18% *Fusarium*, 14% *Cladosporium,* 9% *Mucor*
Erica	<50	<50	<20	*Alternaria*
Lissabon	2.2 × 10^2^	2.1 × 10^2^	<20	84% *Penicillium*, 9% *Alternaria*, 2.5% *Acremonium*, 2.5% *Apergillus*, 2% *Cladosporium*
Mavka	2.6 × 10^2^	2.0 × 10^2^	<100	48% *Alternaria*, 16% *Cladosporium*, 16% *Penicillium*, 7% not recognised, 5% *Mucor*, 5% *Nigrospora*, 3% *Rhizopus*
Merlin	1.4 × 10^2^	1.3 × 10^2^	<10	62% *Alternaria*, 20% *Cladosporium*, 14% *Acremonium*, 4% *Penicillium*
Naya	3.1 × 10^2^	2.2 × 10^2^	<100	67% *Fusarium*, 14.5% *Alternaria*, 14.5% *Cladosporium*, 2% *Aspergillus,* 2% *Eurotium*
Petrina	<20	<20	none	*Alternaria, Cladosporium*
Protina	<50	<20	<50	*Mucor, Penicillium*
Silesia	1.8 × 10^3^	1.7 × 10^3^	<50	61% *Alternaria*, 24% not recognised, 10% *Cladosporium*, 5% *Fusarium*
Sirelia	1.1 × 10^3^	1.0 × 10^3^	<100	82% *Phoma*, 9% *Mucor*, 9% *Penicillium*
Solena	1.3 × 10^2^	<20	1.0 × 10^2^	*Mucor, Penicillium, Thamnidum*

CFU—colony-forming unit.

**Table 7 animals-10-01972-t007:** Apparent total tract digestibility of dry matter and crude protein in control and experimental diets.

Digestibility Coefficients (%)	S0	S5	S10	S15	S20	S25	P
DM	93.91 ± 2.02	93.51 ± 1.91	93.69 ± 2.28	93.90 ± 1.92	94.53 ± 2.59	94.59 ± 2.26	0.17
CP	74.43 ± 0.94	70.63 ± 1.10	70.76 ± 1.27	74.14 ± 0.98	77.28 ± 1.69	76.59 ± 1.88	0.16

S0 groups were offered feed with SBM as the sole protein component in the diet; S5, S10, S15, S20 and S25 of SBM was replaced by soybean seeds (cv. Augusta); P—probability of testing, DM—dry matter, CP—crude protein; results are expressed as means ± standard deviation.

**Table 8 animals-10-01972-t008:** Body weight gain, feed intake and feed conversion ratio parameters during the experimental period.

Parameter	S0	S5	S10	S15	S20	S25	P
1–14 day period							
ADG (kg)	0.38 ^a^ ± 0.05	0.38 ^a^ ± 0.09	0.32 ^a,b^ ± 0.08	0.24 ^cd^ ± 0.07	0.27 ^bc^ ± 0.03	0.18 ^d^ ± 0.04	0.001
ADFI (kg)	0.83 ± 0.11	0.78 ± 0.12	0.77 ± 0.12	0.76 ± 0.10	0.68 ± 0.11	0.66 ± 0.14	0.074
FCR (g/kg)	2.11 ^c^ ± 0.16	2.13 ^c^ ± 0.26	2.50 ^b,c^ ± 0.46	3.16 ^a,b^ ± 0.69	3.34 ^a,b^ ± 0.64	3.72 ^a^ ± 0.91	0.004
15–28 day period							
ADG (kg)	0.67 ^a^ ± 0.07	0.53 ^b^ ± 0.09	0.50 ^bc^ ± 0.08	0.42 ^cd^ ± 0.14	0.36 ^d,e^ ± 0.09	0.31 ^e^ ± 0.08	0.001
ADFI (kg)	1.20 ^a^ ± 0.06	1.13 ^ab^ ± 0.11	1.13 ^ab^ ± 0.06	1.06 ^b^ ± 0.10	1.05 ^b^ ± 0.14	0.94 ^c^ ± 0.14	0.007
FCR (kg/kg)	1.73 ^b^ ± 0.25	2.21 ^b^ ± 0.36	2.06 ^b^ ± 0.27	2.80 ^a^ ± 0.34	3.01 ^a^ ± 0.58	3.18 ^a^ ± 0.58	0.001
1–28 day period							
Final body mass (kg)	24.88 ^a^ ± 1.15	22.63 ^a^^,^^b^ ± 1.44	21.86 ^b^ ± 2.06	19.3 ^c^ ± 1.98	18.25 ^c^ ± 2.44	16.9 ^c^ ± 1.68	0.001
ADG (kg)	0.52 ^a^ ± 0.04	0.45 ^a,b^ ± 0.06	0.42 ^b^ ± 0.06	0.33 ^c^ ± 0.08	0.29 ^c^ ± 0.10	0.24 ^c^ ± 0.06	0.001
ADFI (kg)	0.99 ^a^ ± 0.04	0.96 ^a^ ± 0.09	0.91 ^ab^ ± 0.06	0.88 ^a,b^ ± 0.07	0.87 ^a,b^ ± 0.10	0.80 ^b^ ± 0.06	0.028
FCR (kg/kg)	1.91 ^b^ ± 0.22	2.17 ^b^ ± 0.35	2.21 ^b^ ± 0.20	2.88 ^a^ ± 0.55	2.96 ^a^ ± 0.60	3.37 ^a^ ± 0.66	0.001

S0 groups were offered feed with SBM as the sole protein component in the diet; S5, S10, S15, S20 and S25 of SBM was replaced by soybean seeds (cv. Augusta); ADG—average daily gain, ADFI—average daily feed intake, FCR—feed conversion ratio, P—probability of testing, ^a,b,c,d,e^—differences at *p* < 0.05; results are expressed as means ± standard deviation.

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
