# Peer review of "A Comparison of the Composition and Contamination of Soybean Cultivated in Europe and Limitation of Raw Soy Seed Content in Weaned Pigs’ Diets"

_animals, 2020, doi:10.3390/ani10111972_

Round 1

Reviewer 1 Report

The aim of the present investigation was: 1) to analyse the chemical composition of seeds of soybean varieties available in the EU (Poland); 2) to evaluate the impact of increasing levels of raw soybean seeds on rearing results of piglets; and 3) to determine the effect of increasing levels of raw soybean seeds on the apparent total tract digestibility coefficients of dry matter and crude protein. The goal is very interesting, but the objective 1 seems independent to the aims 2 and 3, as only one soybean variety was tested later. In case you want to link these three goals, you must explain why var. Augusta was used for animal study after screening the nutritive and non-nutritive value of 18 varieties.

In addition, as the nutritive value of seeds is shown only descriptively and the number of seeds is high, perhaps they could be grouped according to some agronomic characteristics (maturity, disease resistance, yield) and establish 2 or 3 new groups containing the average analyses for these cultivars (otherwise the tables are overwhelming). In fact, feed plants cannot control easily the seed cultivar that they receive but only the harvesting time, that normally is associated with differences in crop cultivars cycle. This is also applicable to mycotoxin analyses, that are highly dependent on weather conditions around harvest rather than to a cultivar sensitivity, is it? Discuss. Were the seeds dried after harvesting?  

L18 specify that they were nursery or post-weaning piglets, since the results will depend on pig’s weight

L19 “in which SBM was replaced by 0%, 5%, 10%, 15%, 20% and 25% soybean SEED addition.”

L33-34 which purpose to collect digesta samples? It is not reflected elsewhere.

L158 In seeds, tables are reporting mean and standard error or mean and standard deviation? (for example, in table 2). In the experiment with pigs, the table variables may have tested quadratic responses (6 treatments allow a more detailed evaluation of each response variable!).

Table 1, three decimals are perhaps excessive. Footnote for vitamin-mineral premix supply should express the contents per kg of feed (as the rest of ingredients). This means that

Table 2 and L127, what is the rationale for true protein analysis? Is it possible to have around 10% of non-protein nitrogen?

Table 7, it is a strange that the CTTAD was not affected by soyabean replacement with seed but growth performance was indeed linearly reduced? Was then the metabolic efficiency reduced by soy seed addition? Which was the endpoint of digested amino acids if not used for muscle synthesis? What about faecal consistency or dry matter of faeces, was it affected by soyabean meal replacement by seeds?

Table 8, the standard reporting of growth performance is providing initial body-weight, that must be similar between treatments, final body-weight, average daily gain, average daily feed intake. Why reporting total weight gain and total feed intake? This adds nothing if the reader cannot know the starting point.

In all tables, normally the SEM is reported before the P-value and not thereafter.

Author Response

Dear Reviewer 1,

first of all, we would like to thank for comments and for considering a revised version of our manuscript. All changes made in the manuscript are marked in red. The manuscript in the revised form has been approved by all the co-authors.

  1. The goal is very interesting, but the objective 1 seems independent to the aims 2 and 3, as only one soybean variety was tested later. In case you want to link these three goals, you must explain why var. Augusta was used for animal study after screening the nutritive and non-nutritive value of 18 varieties.

Our response: Thanks for this comment. Firstly, we wanted to know if there are differences among analysed varieties. Soybean cv. Augusta is a model variety in field cultivation and in Poland is still considered as the most suitable for the climatic conditions of Central and Eastern Europe, giving a stable yield in various weather conditions. Moreover, it was created in 2002 at our University. The seeds of this cultivar are characterised by the chemical composition closest to the cultivar's average with the lowest urease activity, which is beneficial for the animals. – It was added into the Material and Methods.

  1. In addition, as the nutritive value of seeds is shown only descriptively and the number of seeds is high, perhaps they could be grouped according to some agronomic characteristics (maturity, disease resistance, yield) and establish 2 or 3 new groups containing the average analyses for these cultivars (otherwise the tables are overwhelming). In fact, feed plants cannot control easily the seed cultivar that they receive but only the harvesting time, that normally is associated with differences in crop cultivars cycle. This is also applicable to mycotoxin analyses, that are highly dependent on weather conditions around harvest rather than to a cultivar sensitivity, is it? Discuss.

Our response: As we indicated, the aim of the present investigation was to analyse the chemical composition of seeds of different soybean varieties available in the EU (Poland), but not focusing on agronomic characteristics. For us more important is the nutritional value of soybeans growing in the EU. This study was realized within the National program and work package 4 -"Improvement of native plant protein feeds utilization in animal feed"- part 4.1 -Monitoring of nutrients and anti-nutritional substances of new varieties of legume seeds and other domestic sources of protein feeds in terms of their nutritional suitability. Besides, the highly variable humidity and thermal conditions that occurred in the studied growing seasons (especially in May and June in Poland) could significantly affect the contents of nutrients and anti-nutrients, regardless of the soybean variety. Therefore, we do not find it necessary to divide varieties according to a different key. It was not changed because the other Reviewers did not mention this point.

  1. Were the seeds dried after harvesting?  

Our response: As indicated in the section M&M (L. 90-92), soy seeds came from different plant stations and seed drying was performed probably when the dry matter content of the harvested crop was below 86%. We do not know if/which seeds required drying.

  1. L18 specify that they were nursery or post-weaning piglets, since the results will depend on pig’s weight –

Our response:  changed “pigs” to “post-weaning piglets”. L. 19-20.

  1. L19 “in which SBM was replaced by 0%, 5%, 10%, 15%, 20% and 25% soybean SEED addition.”

Our response: in L. 21 added “seeds

  1. L33-34 which purpose to collect digesta samples? It is not reflected elsewhere.

Our response: We agree with the Reviewer. In the last 3 days of the experiment the samples of excreta from each animal separately were collected three times per day. L. 36-37

  1. L158 In seeds, tables are reporting mean and standard error or mean and standard deviation? (for example, in table 2). In the experiment with pigs, the table variables may have tested quadratic responses (6 treatments allow a more detailed evaluation of each response variable!).

Our response: All the data are shown as mean values and SD – Tables 7 and 8. Concerning the experiment on pigs – we used a standard procedure and a prerequisite when statistically evaluating biological data. It is one of the most powerful statistical tools testing whether a normal distribution adequately describes a set of data. The other Reviewers did not mention this point.

  1. Table 1, three decimals are perhaps excessive.

Our response: We agree. We changed to two decimals.

  1. Footnote for vitamin-mineral premix supply should express the contents per kg of feed (as the rest of ingredients).

Our response: We followed this suggestion and changed mineral and vitamin premix contents per 1 kg feed- L. 134-138.

  1. Table 2 and L127, what is the rationale for true protein analysis? Is it possible to have around 10% of non-protein nitrogen?

Our response: Legumes (including soybean seeds) contain high concentrations of protein. However, when evaluating them as a possible food source for animals it is necessary to determine true protein content, because many legumes also contain substantial concentrations of non-protein amino acids and other N-containing substances, which contribute significantly to a total nitrogen measurement. For example, the study of Lucas et al. (1988) showed that about 33% of the total nitrogen was in non-protein nitrogen form. Some non-protein amino acids have undesirable physiological effects on the animals which consume them. Bell (2003) also found it necessary to learn more about non-protein amino acids in feed components because of their importance in animal nutrition.

Lucas, B., Guerrero, A., Sigales, L., & Sotelo, A. (1988). True protein content and non-protein amino acids present in legumes seeds. Nutrition reports international (USA).

Bell, E. Arthur. (2003). Nonprotein amino acids of plants: significance in medicine, nutrition, and agriculture." Journal of agricultural and food chemistry 51.10, 2854-2865.

  1. Table 7, it is a strange that the CTTAD was not affected by soyabean replacement with seed but growth performance was indeed L.arly reduced? Was then the metabolic efficiency reduced by soy seed addition? Which was the endpoint of digested amino acids if not used for muscle synthesis? What about faecal consistency or dry matter of faeces, was it affected by soyabean meal replacement by seeds?

Our response: We did not find any anomaly and variations in the consistency and dry matter contents in the faeces of pigs from all the groups. We think that the reduced growth of piglets was connected with a lower feed intake and with feed palatability rather than their digestibility. Moreover, we analysed only dry matter and protein digestibility of feed and, in order to verify the results the part of samples was reanalysed, but the results were the same. Besides, in the accordance with the Reviewer’s indication it is possible that the metabolic efficiency of feed containing more raw soy seeds was reduced, but we did not analyse energy in this research.

  1. Table 8, the standard reporting of growth performance is providing initial body-weight, that must be similar between treatments, final body-weight, average daily gain, average daily feed intake. Why reporting total weight gain and total feed intake? This adds nothing if the reader cannot know the starting point.

Our response: We agree. In table 8 we deleted these table parts and added final body mass.

  1. In all tables, normally the SEM is reported before the P-value and not thereafter.

Our response: It was changed because SEM was removed from Tables 7 and 8.

English was revised by a professional language editor.

We hope that the changes made have improved the quality of the manuscript. We would be pleased if the revised manuscript could be published in MDPI Animals.

Yours sincerely,

Małgorzata Kasprowicz-Potocka

Reviewer 2 Report

General comment

The study aimed to characterized a large variety of soy seed in terms of bramotological analysis and in term of safety and antinutitional properties. In addition, the manuscript explores the effect of dietary inclusion of soy seed in piglets.

The topic is interesting; some important information is missing and the methodology is lacking on some essential detail and doubts are raised on the methodology adopted. The discussion part should be deeply revised, some aspect of the discussion is not clear or are not well discussed.

English language should be revised by professional reviewer, there are grammatical errors and the construction of the phrases sometimes are not clear.

Simple Summary

The simple summary should be modified accordingly to authors guidelines. It is not an Abstract. 

Line 14: I suggest to modify this phrase "Soya is the most popular feed crop in the world", should be better appropriated, maybe authors intended " is the major source of protein"?

Please replace Soya with Soy for the whole manuscript.

From Instructions for Authors:

“Simple Summary: It is vitally important that scientists are able to describe their work simply and concisely to the public, especially in an open-access on-line journal. The simple summary consists of no more than 200 words in one paragraph and contains a clear statement of the problem addressed, the aims and objectives, pertinent results, conclusions from the study and how they will be valuable to society. This should be written for a lay audience, i.e., no technical terms without explanations. No references are cited and no abbreviations. Submissions without a simple summary will be returned directly. Example could be found at https://www.mdpi.com/2076-2615/6/6/40/htm.”

Abstract

The abstract is lacking of some important information related to the methodology adopted in this study.

Line 27: ""specific chemical composition" please be more consistent. The aim of the study seems not in line with analysis and experimental trial carried out. Why authors adopted raw seeds? it is well known the antinutritional property of soy without any thermal treatment. Please better specify in the aim of the study why the experimental trial used raw seed.

Line 31-32: Please, specify marker method. the duration of the trial is missing.

Line 32-33: please specify all the percentage of soy used in the trial

Line 34-36: all the bromatolociacl and "toxicological" analysis should me mentioned before.

Line 39-40: specify "performance parameter", what did you measured? body weight? feed intake? be more consistent

Introduction

Line 44-53: the first part of the introduction is not appropriate. The introduction should explain and give an overview of the "issue". Thus, the first part of the introduction seems do not follow the second part.

I highly suggest to focus on the different variety of soy and the related differences among varieties. The reader cannot understand what could be the differences among varieties and what is the real problem using the raw seed especially in monogastric (I suggest to spend few words on this part) 

Line 55-57: Please, add citations.

Line 57-58: "technological groups", it is not clear., please better specify.

Line 59-61: The sentence is not fully correct. it is true that environmental condition, soy varieties influence the chemical composition of soybean. However, soybean seeds (raw) is not an "usual" feed ingredient, firstly because the nutritional composition of soybean seed lead to prefer to adopt soybean meal instead due to the antinutitional properties; secondly, the market offers a large variety of sot by-product which are thermal treatment reducing the antinutitional properties of soy. 

Materials and methods

Line 87. Please, specify the crossbreed. Add body weight standard deviation. Add the age of piglets

Line 88-89: please revise English. "individuals"?

Line 91: please specify where did you purchase the soybean seeds

Line 92-93: did you buy the feed? add more information.

Line 96: please use day o, day 14 and day 28 instead.

How did you weight the animals? individually? please specify? how did you recored feed intake? daily? weekly? specify it.

Table 1: Please use 2 or 1 digits after comma. Numbers should be uniformed for the entire manuscript

Line 105-119: this section is lacking of information.

How authors calculated the apparent total tract digestibility if animals were housed in a common pen?

it is possible calculate ATTD if animals are allotted in a metabolic cage? 

There are no citations about ATTD formula adopted.

The methodology is lacking, and is not sufficient to conduct this type of experimental trial.

Line 121-135:

- In chemical analysis please specify the type of sample analyzed (feed, faces etc..)

- the methods reported are not well reported.

- line 133: described by (14) please add the name of the author and briefly describe the method as well as for TIA and titanium dioxide.

line 136-144: please better describe the methodology, more in formation are needed concerning the extraction and purification (such as the concentration of reagents used for the extraction...). Better describe the method adopted for ZEA.

Line 158-162: it is not clear what means "single records" better specify. 

it is not reported if the statistical model used were specified the fixed effect of tretmant, experimental day or interaction between treatment and experimental day. is it not reported if the data wll be presented ad mean or LSMEAN, standard deviation or standard error or SEM. 

Revise English of this section. 

Results

Line 167: nutrients were recalculated? maybe were expressed as % of DM. 

Cv. was not specified before

Line 187: "Urease was measured only in some sample" this aspect was not mentioned before. Specify the number of sample

Line 200-201: please specify the acronyms. 

Line 204: ....but their concentrations were low.... it was low compare to the maximum admited level? please specify 

Table 4: the title of table 4 should be revised., what is total g/kg??? 

Table 5: this is a huge table; the reader get lost. 

animal experiment section page 23: is not reported le number of lines

... SBM was supplemented at 5% and 10%,... did you supplement soybean meal? please modify the sentence,

page 24:

table 8: all the acronyms were not reported before, please add this part in M&M section.

Modify DWG-daily weighs gain with average daily gain which is common used as well as for DFI with average daily feed intake-ADFI

Report body weight, it in an essential parameter.

I do not understand why report total weight gain and total feed intake; these data do not show any adding value. please report only average daily gain- ADG and average daily feed intake-ADFI

How did you calculate the period 1-14, 15-28 etcc in the calculation was included the day 14 or not?

should be better use 1-14, 14-28, etc…

How did you calculate dry matter and crude protein digestibility if animals wer not allotted in a metabolic cages? i can undersand that the experimental trial adopted a marker, however without the amount of urine and feaces producted by the single animal it is not possible calculate these parameters. You can calculate total tract digestibility of nutrients but not of dry matter.

See;

Dewen Liu , Hu Liu , Defa Li & Fenglai Wang (2019) Determination of nutrient digestibility in corn and soybean meal using the direct and substitution methods as well as different basal diets fed to growing pigs, Journal of Applied Animal Research, 47:1, 184-188, DOI: 10.1080/09712119.2019.1597725

Wenxuan Dong , Qiuyun Wang , Jianyu Chen , Ling Liu & Shuai Zhang (2019) Apparent total tract digestibility of nutrients and the digestible and metabolizable energy values of five unconventional feedstuffs fed to growing pigs, Journal of Applied Animal Research, 47:1, 273-279, DOI: 10.1080/09712119.2019.1625778

Page 25: why add information about organic cultivation? did you have organic soy in the sample set?

Please, revise English   ...of an opinion that...

Discussion- Animal experiment section

Authors mainly focused their attention on apparent digestibility, however zootechnical performance were not exhaustively discussed. The animal experiment section is not well discussed, authors did not discuss the main results about zootechnical performance, which were compromised by saw soy seed supplementation.

Data showed that nutrient digestibility was not modulated by soy seed, however the inclusion of different concentration of soy seed reduced body weight and feed intake of piglets. Some considerations should be done in light of the reduction of zootecnical performance, probably due to antinutitional properties of raw soy which have reduced the utilization of some essential nutrients (I suppose). 

The discussion part is lacking, the author did not well discuss the data and focalized their attention only on   chemical composition of soy. 

Conclusion section

"More seeds also contained mycotoxins, especially zearalenone and deoxynivalenol. " authors should specify if this aspect could compromize the use ad feed ingredient and the animal health, addig the legislative limits. i suggest to modulate the phrase "Diets with soybean seeds were characterised by a high digestibility of dry matter and protein regardless of the proportion of raw seeds in the feed " i suggest to highlight that digestibility was no modulated.

The sentence " Replacement of soybean meal with raw soybean seeds in the diet for young pigs in the amount of 5% had no negative effect on the animals’ performance, but higher raw seed levels resulted in a lower feed intake and utilization as well as reduced growth parameters" is highly speculative and contradictory. animal performance was lower!!!

the conclusion should be deeply revise

Author Response

Dear Reviewer 2

first of all, we would like to thank for comments and for considering a revised version of our manuscript. All changes made in the manuscript are marked in red. The manuscript in the revised form has been approved by all the co-authors.

  1. Simple Summary. The simple summary should be modified accordingly to authors guideL.s. It is not an Abstract. 

Our response: Thanks for this comment.  The "Simple Summary" was revised once more and adjustments were made according to the guide L.s.

  1. 14: I suggest to modify this phrase "Soya is the most popular feed crop in the world", should be better appropriated, maybe authors intended " is the major source of protein"?

Our response: We followed this suggestion and corrected: Soy is the major source of protein in animal feeds worldwide. - L. 14

  1. Please replace Soya with Soy for the whole manuscript.

Our response: We followed this suggestion and changed in the whole manuscript.

  1. The abstract is lacking of some important information related to the methodology adopted in this study.

Our response:  We agree with the Reviewer. Nevertheless, we can`t provide more information (in the Abstract) related to the methodology adopted in this study, because, in accordance with the journal's guide L.s, the abstract should have about 200 words. Now the abstract has 216 words. Moreover, the other Reviewers did not notice any significant gaps in information in the Abstract. Only some details were added according to the remark by Reviewer 2.

  1. 27: ""specific chemical composition" please be more consistent. The aim of the study seems not in L. with analysis and experimental trial carried out. Why authors adopted raw seeds? it is well known the antinutritional property of soy without any thermal treatment. Please better specify in the aim of the study why the experimental trial used raw seed.

Our response: “Specific” was removed. L 29.

  1. 31-32: Please, specify marker method. the duration of the trial is missing.

Our response: Marker method was added - L 35. Trial duration was 28 d - L. 34.

  1. 32-33: please specify all the percentage of soy used in the trial.

Our response:  it was added – L. 35-36.

  1. 34-36: all the bromatolociacl and "toxicological" analysis should me mentioned before.

Our response: It is not possible to add it in this section.

  1. 39-40: specify "performance parameter", what did you measured? body weight? feed intake? be more consistent

Our response: It was added in L. 32-33. (body weight gain, feed intake and feed utilization).

  1. I L. 44-53: the first part of the introduction is not appropriate. The introduction should explain and give an overview of the "issue". Thus, the first part of the introduction seems do not follow the second part.I highly suggest to focus on the different variety of soy and the related differences among varieties. The reader cannot understand what could be the differences among varieties and what is the real problem using the raw seed especially in monogastric (I suggest to spend few words on this part) .

Our response: It was changed according to the remark by Reviewer 2. L. 49-66.

  1. 55-57: Please, add citations.

Our response: the citation was added [5, 6] (L 69).

  1. 57-58: "technological groups", it is not clear., please better specify.

Our response: Some details were added (L 70).

  1. 59-61: The sentence is not fully correct. it is true that environmental condition, soy varieties influence the chemical composition of soybean. However, soybean seeds (raw) is not an "usual" feed ingredient, firstly because the nutritional composition of soybean seed lead to prefer to adopt soybean meal instead due to the antinutitional properties; secondly, the market offers a large variety of sot by-product which are thermal treatment reducing the antinutitional properties of soy. 

Our response: We agree with the Reviewer’s opinion and this sentence was deleted. (L 71-73).

  1. 87. Please, specify the crossbreed. Add body weight standard deviation. Add the age of piglets

Our response: We followed this suggestion and crossbreed, body weight standard deviation and age of piglets were added. L. 113

  1. 88-89: please revise English. "individuals"?

Our response: This part was restructured and some details were added. “Individuals” was removed. L. 114-117.

  1. 91: please specify where did you purchase the soybean seeds

Our response: We followed this suggestion. Some details were added. L. 94-100.

  1. 92-93: did you buy the feed? add more information.

Our response: The information about feed preparation was added. L 124-125.

  1. 96: please use day o, day 14 and day 28 instead.

Our response: It was done. L. 129.

  1. How did you weight the animals? individually? please specify? how did you recored feed intake? daily? weekly? specify it.

Our response: All details were added in the text. L. 125-127.

  1. Table 1: Please use 2 or 1 digits after comma. Numbers should be uniformed for the entire manuscript

Our response: We followed this suggestion and used 2 digits after the decimal point uniformly for the entire manuscript.

  1. 105-119: this section is lacking of information.

Our response: It was improved.

  1. How authors calculated the apparent total tract digestibility if animals were housed in a common pen?

Our response: Pigs were randomly allocated to six dietary treatments (of 8 animals per group) and were kept in individual cages. It was added in text. L. 114-115.

  1. it is possible calculate ATTD if animals are allotted in a metabolic cage? 

Our response: It is the index method described briefly by Adeola (2001) B. Index method. and also used in our previous research.

Kasprowicz-Potocka M., Zaworska, A., Kaczmarek S. A., Rutkowski A. (2016). The nutritional value of narrow-leafed lupine (Lupinus angustifolius) for fattening pigs. Archives of Animal Nutrition, 70(3), 209-223.

  1. There are no citations about ATTD formula adopted.

Our response: A citation was added – Adeola, 2001. L. 147.

The methodology is lacking, and is not sufficient to conduct this type of experimental trial.

Our response: It was a mistake. The correct equation was added according to Adeola, 2001 – Eq. (40.2). L. 149-150.

  1. 121-135:- In chemical analysis please specify the type of sample analyzed (feed, faces etc..)

Our response: We agree with the Reviewer and some details were added (L. 195)

  1. the methods reported are not well reported.- L. 133: described by (14) please add the name of the author and briefly describe the method as well as for TIA and titanium dioxide.

Our response: As recommended, the authors described the indicated methods. L. 172-200.

  1. 136-144: please better describe the methodology, more in formation are needed concerning the extraction and purification (such as the concentration of reagents used for the extraction...). Better describe the method adopted for ZEA.

Our response:  As recommended, the authors described the methodologies in detail. L. 202-268.

  1. 158-162: it is not clear what means "single records" better specify. it is not reported if the statistical model used were specified the fixed effect of tretmant, experimental day or interaction between treatment and experimental day. is it not reported if the data wll be presented ad mean or LSMEAN, standard deviation or standard error or SEM. Revise English of this section. 

Our response: We agree with the Reviewer in this regard and changed the part of the section. Additionally, we added a sentence: The obtained results were analysed statistically by calculating the arithmetic mean and ± SD for each characteristic. L. 290-296. English was improved.

  1. Results
  2. 167: nutrients were recalculated? maybe were expressed as % of DM. 

Our response: Thanks for this comment. Nutrients were expressed as % of DM. L.301.

  1. was not specified before

Our response: We added the full name – “cultivar”. L. 82.

  1. 187: "Urease was measured only in some sample" this aspect was not mentioned before. Specify the number of sample

Our response: We agree with the Reviewer in this regard and added in the M & M section the number of samples – L. 190-194.

  1. 200-201: please specify the acronyms. 

Our response: Full names were added in the M&M section. L.247-248.

  1. 204: ....but their concentrations were low.... it was low compare to the maximum admited level? please specify 

Our response: We agree with the Reviewer and an explanation was added in the text. L. 338-339.

  1. Table 4: the title of table 4 should be revised., what is total g/kg??? 

Our response: The title of Table 4 and the phrase in columns were changed: Total content RFOs (g/kg); % of sugars in RFO`s (L. 359)

  1. Table 5: this is a huge table; the reader get lost. 

Our response: We agree with the Reviewer, but in our opinion it is hard to present these data in any other way.

  1. animal experiment section page 23: is not reported le number of L.s

Our response: It was improved.

  1. ... SBM was supplemented at 5% and 10%,... did you supplement soybean meal? please modify the sentence,

Our response: thank you for your critical observation. We agree and it was changed (L. 371).

  1. page 24:table 8: all the acronyms were not reported before, please add this part in M&M section. Modify DWG-daily weighs gain with average daily gain which is common used as well as for DFI with average daily feed intake-ADFI

Our response: Full names were added in the M&M section (L. 127-128) and it was improved in other places.

  1. Report body weight, it in an essential parameter.I do not understand why report total weight gain and total feed intake; these data do not show any adding value. please report only average daily gain- ADG and average daily feed intake-ADFI

Our response: We agree. We deleted these values and only final body mass was added. (Table 8).

  1. How did you calculate the period 1-14, 15-28 etcc in the calculation was included the day 14 or not?should be better use 1-14, 14-28, etc…

Our response: We calculated periods: 1-14 and next 15-28. Information about periods is given in Table 8.

  1. How did you calculate dry matter and crude protein digestibility if animals wer not allotted in a metabolic cages? i can undersand that the experimental trial adopted a marker, however without the amount of urine and feacesproducted by the single animal it is not possible calculate these parameters. You can calculate total tract digestibility of nutrients but not of dry matter.See;

DewenLiu , Hu Liu , Defa Li &Fenglai Wang (2019) Determination of nutrient digestibility in corn and soybean meal using the direct and substitution methods as well as different basal diets fed to growing pigs, Journal of Applied Animal Research, 47:1, 184-188, DOI: 10.1080/09712119.2019.1597725

WenxuanDong ,Qiuyun Wang , Jianyu Chen , Ling Liu & Shuai Zhang (2019) Apparent total tract digestibility of nutrients and the digestible and metabolizable energy values of five unconventional feedstuffs fed to growing pigs, Journal of Applied Animal Research, 47:1, 273-279, DOI: 10.1080/09712119.2019.1625778

Our response: As we pointed out before, we used a standard method described by Adeola (2001) (see note 22).

  1. Page 25: why add information about organic cultivation? did you have organic soy in the sample set?Please, revise English   ...of an opinion that...

Our response: We agree. Organic soy cultivation data were removed. L. 401.

  1. Discussion- Animal experiment section Authors mainly focused their attention on apparent digestibility, however zootechnical performance were not exhaustively The animal experiment section is not well discussed, authors did not discuss the main results about zootechnical performance, which were compromised by saw soy seed supplementation.Data showed that nutrient digestibility was not modulated by soy seed, however the inclusion of different concentration of soy seed reduced body weight and feed intake of piglets. Some considerations should be done in light of the reduction of zootecnical performance, probably due to antinutitional properties of raw soy which have reduced the utilization of some essential nutrients (I suppose). The discussion part is lacking, the author did not well discuss the data and focalized their attention only on chemical composition of soy. 

Our response: We concentrated on digestibility results because they aren`t in the accordance with our hypothesis, so we tried to understand this phenomenon. The performance results are rather not surprising. Besides, there is the lack of new data about the use of raw soy seeds in diets for pigs to compare [38]. We added some considerations in the text and suitable literature sources [39, 40]. L. 475-485.

  1. Conclusion section. "More seeds also contained mycotoxins, especially zearalenone and deoxynivalenol. " authors should specify if this aspect could compromize the use ad feed ingredient and the animal health, addig the legislative limits. i suggest to modulate the phrase "Diets with soybean seeds were characterised by a high digestibility of dry matter and protein regardless of the proportion of raw seeds in the feed " i suggest to highlight that digestibility was no modulated.The sentence " Replacement of soybean meal with raw soybean seeds in the diet for young pigs in the amount of 5% had no negative effect on the animals’ performance, but higher raw seed levels resulted in a lower feed intake and utilization as well as reduced growth parameters" is highly speculative and contradictory. animal performance was lower!!!

Our response: All the suggestions of the Reviewer were accepted and changes were made in the text. L. 490-497.  In the case of 5% replacement no negative effect of the diet on pigs` performance was found (Table 8).

English was revised by a professional language editor.

Submission Date

31 August 2020

Date of this review

09 Sep 2020 09:16:32

We hope that the changes made have improved the quality of the manuscript. We would be pleased if the revised manuscript could be published in MDPI Animals.

Yours sincerely,

Małgorzata Kasprowicz-Potocka

Reviewer 3 Report

The work is interesting and includes many detailed chemical analyzes of non-GMO soybeans. There was no research hypothesis, e.g. whether and in what amount can SBM be replaced with non-GMO soybeans? The material and methods should be specified. Details in the text and below.

Line 19 added soybean meal (SBM)
Line 31 added crude protein
Line 42 piglets
Line 53 soybean meal
Line 64 There was no research hypothesis, e.g. whether and in what amount can SBM be replaced with non-GMO soybeans?
Line 72-79, It is a pity that the soybean seeds of different cultivars were not obtained from the same soil-climatic, agrotechnical and comparable conditions (years). Therefore, it is difficult to conclude that the determinations of chemical composition and nutritional value refer to cultivars. Please underline this clearly in the discussion.
Line 87 please provide sex ratio and possible gilts and barrows
Line 92 Rapeseed oil was added to the experimental diets to make diets iscaloric.
Line 106 Please describe how the faeces were collected, directly from the anus or from the straw floor in pens?
Line 132-135, please specify (describe briefly) the methods for the determination of raffinose, phytate-P, TIA, and titanium dioxide and the equipment used (company, country).

Author Response

Dear Reviewer 3,

first of all, we would like to thank for comments and for considering a revised version of our manuscript. All changes made in the manuscript are marked in red. The manuscript in the revised form has been approved by all the co-authors.

  1. L. 19 added soybean meal (SBM)-

Our response: We followed this suggestion and added soybean meal (L. 20)

  1. 31 added crude protein

Our response: in L. 34 we added “crude”

  1. L. 42 piglets

Our response: in L. 47 we changed “pigs” to “piglets”

  1. 53 soybean meal

Our response: This part was changed according to the remark by Reviewer 2.

  1. L. 64 There was no research hypothesis, e.g. whether and in what amount can SBM be replaced with non-GMO soybeans?

Our response: The hypothesis was added (74-77).

  1. L. 72-79, It is a pity that the soybean seeds of different cultivars were not obtained from the same soil-climatic, agrotechnical and comparable conditions (years). Therefore, it is difficult to conclude that the determinations of chemical composition and nutritional value refer to cultivars. Please underL. this clearly in the discussion.

It is not possible to obtain all these varieties (seeds) from the same conditions because they are designed to different environments (soil, climate) and they have also different maturation characteristic. As was suggested by the Reviewer, some comments were added in the Discussion (L. 417-421).

  1. L. 87 please provide sex ratio and possible gilts and barrows

Our response: We followed this suggestion and added “barrows” in L. 112.

  1. L. 92 Rapeseed oil was added to the experimental diets to make diets iscaloric.

Our response: We added the above sentence (L. 121-122).

  1. L. 106 Please describe how the faeces were collected, directly from the anus or from the straw floor in pens?

Our response: We added in L.- 143-144. The fresh faeces were collected from the straw floor in pens and the unwanted straw waste was removed.

  1. L. 132-135, please specify (describe briefly) the methods for the determination of raffinose, phytate-P, TIA, and titanium dioxide and the equipment used (company, country)

Our response: As recommended, the authors briefly described the indicated methods. L. 172-200.

Submission Date

31 August 2020

Date of this review

09 Sep 2020 09:16:32

English was revised by a professional language editor.

We hope that the changes made have improved the quality of the manuscript. We would be pleased if the revised manuscript could be published in MDPI Animals.

Yours sincerely,

Małgorzata Kasprowicz-Potocka

Correspondig author

Round 2

Reviewer 1 Report

The authors addressed most of the reviewers' concerns.

However, I keep on considering that the tables are overwhelming and does not fit journal publishing standards and reader appeal. You replied in L371-374 that "It should be emphasised here that the analysed soybean varieties were not obtained from the same soil-climatic, agritechnical and other comparable conditions (years), therefore it is difficult to definitely state that the assayed chemical composition and nutritional value are strictly cultivar-dependent." Therefore, it would be necessary to summarize Table outcomes by grouping the cultivars according to feed nutrient content (for example, establishing two-three groups according to CP content of the raw seeds, or two-three groups based on anti-nutrient content, and thereby reducing the data in tables).

Finally, regarding the animal experiment, the authors must emphasize in the discussion that they hypothesize that the reduced growth of piglets may be related to a lower feed intake due to reduced palatability rather than to a decreased nutrient digestibility.

Reviewer 2 Report

The authors substantially improved the manuscript following previous suggestion. Minor changes are needed. 

Line 22: (but to varying degrees), what authors means? is is not clear maybe adopt other therms

Line 46: "concern concerning" better "related to "

Line 62: i suggest to avoid " in our opinion", and add few information why raw seeds may decrease nutrient digestibility, thus the antinutritional properties. please add also referencese.

Line 63: "thaim" add space

Line 99-100: i suggest to use another way to cite the legislation

Line 307: please correct NDF, is " Neutral detergent fiber (NDF
